# Regulation of the Epithelial to Mesenchymal Transition in Osteosarcoma

**DOI:** 10.3390/biom13020398

**Published:** 2023-02-20

**Authors:** Kristin Hinton, Andrew Kirk, Paulose Paul, Sujata Persad

**Affiliations:** 1Division of Orthopaedic Surgery, Department of Surgery, University of Alberta, Edmonton, AB T6G 2B7, Canada; 2Faculty of Medicine and Dentistry, University of Alberta, Edmonton, AB T6G 2R7, Canada; 3Department of Pediatrics, University of Alberta, Edmonton, AB T6G 2R3, Canada

**Keywords:** osteosarcoma, epithelial-mesenchymal transition, transcriptional regulation, long non-coding RNAs, circular RNAs, microRNAs, tumor microenvironment, cancer therapeutics

## Abstract

The epithelial to mesenchymal transition (EMT) is a cellular process that has been linked to the promotion of aggressive cellular features in many cancer types. It is characterized by the loss of the epithelial cell phenotype and a shift to a more mesenchymal phenotype and is accompanied by an associated change in cell markers. EMT is highly complex and regulated via multiple signaling pathways. While the importance of EMT is classically described for carcinomas—cancers of epithelial origin—it has also been clearly demonstrated in non-epithelial cancers, including osteosarcoma (OS), a primary bone cancer predominantly affecting children and young adults. Recent studies examining EMT in OS have highlighted regulatory roles for multiple proteins, non-coding nucleic acids, and components of the tumor micro-environment. This review serves to summarize these experimental findings, identify key families of regulatory molecules, and identify potential therapeutic targets specific to the EMT process in OS.

## 1. Introduction

Osteosarcoma (OS) is a primary bone malignancy with an annual incidence of 2–4 per million [1]. It typically affects children, teens, and young adults [2], with a peak incidence from ages 10–19 [1], a second peak in adults over 60 [2], and a slight male preponderance [3]. The overall 5-year survival rate for OS is 60% but decreases to 27% in the presence of distant metastases [4]; the rate of metastases at diagnosis is 18% [5]. 

The origin of OS is poorly understood. As a sarcoma, it arises from mesenchymal cells, but it is not currently known whether the precursor cells are osteoblasts or mesenchymal stem cells [6]. Although the etiology of OS is largely a mystery, multiple risk factors have been identified. These include medical conditions such as hereditary retinoblastoma, Li-Fraumeni syndrome, Werner syndrome, Rothmund-Thompson syndrome, Bloom syndrome, and Paget’s disease [3]. Other risk factors include exposure to ionizing radiation and alkylating agents, both of which may have been used in the treatment of a prior malignancy [3].

The mainstay of treatment for osteosarcoma is surgical resection and frequently involves both neoadjuvant and adjuvant chemotherapy for higher grade tumors [7]. While advances in surgical techniques and chemotherapeutic regimens were associated with an initial improvement in outcomes, overall survival in OS has not significantly changed in several decades [8]. As medicine becomes more personalized, there is a growing interest in the identification of novel targeted therapies. A key component in developing targeted therapy is identifying specific pathways, proteins, or other molecules essential to cancer cell function. One of the cellular features often associated with aggressive cancers is the epithelial to mesenchymal transition (EMT). 

## 2. EMT in Cancer

EMT is depicted in Figure 1. It is a process by which cells exhibiting an epithelial phenotype adopt a mesenchymal phenotype, which facilitates migration, invasion, and metastasis [9]. It exists in equilibrium with a reverse and complementary process, the mesenchymal to epithelial transition (MET), wherein cells revert back to an epithelial phenotype. Primary epithelial tumors exhibit epithelial cell markers such as E-cadherin. These cells demonstrate apical polarity, adhesion to a basement membrane, and tight cellular junctions [10]. For many cancers, EMT is critical in the early transition from normal to malignant cells. It is characterized by downregulation of epithelial cell markers, destabilization and loss of cell–cell junctions, loss of adherence to basement membrane and apical polarity, and cytoskeletal reorganization [9]. The result of these changes is a cell with mesenchymal morphology and characteristics.

Given the migratory potential of mesenchymal cells compared to epithelial cells, EMT has long been linked to cancer metastasis. However, inhibition of EMT has not been shown to affect the establishment of cancer metastases in vivo [11,12], and the cells found within metastatic tumors are more likely to exhibit an epithelial phenotype [12,13]. Despite this, tumor cells that have undergone EMT appear to drive local invasion and angiogenesis of the primary tumor [13]. These results suggest that EMT is critical for tumor invasion into the local vascular system, allowing cells to migrate to distant organs where secondary tumors are established largely by cells with an epithelial phenotype, which have a greater propensity for proliferation [9]. These may be either cells that have undergone EMT and subsequently MET or primary tumor cells that did not undergo EMT [13].

The molecular pathways associated with EMT are summarized in Figure 2. Zinc-finger E-box binding homeobox (ZEB), snail family transcriptional repressor 1 (SNAIL), snail family transcriptional repressor 2 (SLUG), and twist-related protein (TWIST) are well-known EMT transcription factors that are established downstream targets of multiple signaling pathways, including the canonical *wnt*/β-catenin pathway, the neurogenic locus notch homolog protein (Notch) pathway, the Transforming Growth Factor β/Suppressor of Mothers Against Decapentaplegic (TGFβ/SMAD) pathway, the phosphoinositide 3-kinase (PI3K)/Akt pathway, the mitogen-activated protein kinase (MAPK) pathway, the Ras/Raf/Mitogen-activated protein kinase/ERK kinase/extracellular-signal-regulated kinase (RAS/RAF/MEK/ERK) axis, and the Janus kinase-Signal Transducer and Activator of Transcription JAK/STAT pathway [10]. These signaling cascades often interact, share many intermediaries, and impact the regulation of one another. This presents a challenge for studying and targeting EMT, as the individual pathways are difficult to isolate. 

## 3. EMT Signaling Pathways

### 3.1. TGFβ/SMAD Pathway

The TGFβ family of proteins includes three TGFβ isoforms, activins, and bone morphogenic proteins (BMPs) [14]. In EMT, TGFβs bind to TGFβ receptors (1/2), which initiate a signaling cascade, leading to the increased transcription of genes involved in EMT. Binding of TGFβ to its receptors (1/2) leads to phosphorylation of SMAD2 and SMAD3, which then form a complex with SMAD4. BMPs also bind TGFβ receptors, activating SMAD1 and SMAD5 and then forming a complex with SMAD4. These trimeric complexes migrate to the nucleus to act as transcription factors.

SMAD complexes activate the mesenchymal genes vimentin and fibronectin, as well as the EMT transcription factors Snail, Slug, Zinc finger E-box-binding homeobox 1 (ZEB1) and Twist. These, in turn, repress E-cadherin and can upregulate the expression of TGFβ ligands, establishing a positive feedback loop to maintain EMT [9,15,16].

### 3.2. Canonical wnt Pathway

The canonical *wnt* pathway is considered to be a key activator of EMT [9]. Signaling is initiated by a group of *wnt* ligands that bind to Frizzled receptors and trigger a cascade of events, leading to the nuclear translocation of β-catenin. β-catenin is constitutively produced in the cell and stored in cytosolic pools. In the absence of *wnt* signaling, phosphorylated β-catenin is associated with a destruction complex, ubiquitinated, and degraded by proteasomes. Following activation of the canonical *wnt* pathway, β-catenin is dephosphorylated and translocates to the nucleus, where it acts as a transcriptional co-factor to induce the expression of genes involved in cell differentiation, proliferation, and tumorigenesis [17,18]. 

This pathway has been directly implicated in EMT via the expression of Twist, Slug, N-cadherin, and the repression of E-cadherin [19]. The known EMT transcription factor Snail has been shown to positively regulate *wnt* signaling [20]. The inhibition of Secreted Frizzled Related Protein 1 (SFRP), a negative regulator of *wnt* ligands, has also been shown to have EMT-like effects in breast carcinoma cells in vitro, while sensitizing them to TGFβ-induced EMT [21].

β-catenin is also located at the cell membrane as part of an E-cadherin-containing multi-component adherens junction complex, which is a component of cell–cell interaction junctions. β-catenin contributes to anchoring E-cadherin, a transmembrane cell–cell adhesion protein at the cell surface to the intracellular actin cytoskeleton. β-catenin is released from the adherens complex upon disruption of these adherens junctions between cells. Once available in the cytosol, it enters the pathway described above and is either phosphorylated and degraded or, if the *wnt* pathway is active, dephosphorylated and translocated to the nucleus to function as a transcription factor for EMT-genes [22]. E-cadherin can therefore act as a negative regulator of the canonical *wnt* pathway by sequestering most of the β-catenin in the epithelial cell membrane.

### 3.3. Notch Pathway

The Notch pathway has been implicated in inducing EMT in both normal and neoplastic tissues, and is involved in controlling cell fate, differentiation, and proliferation. Four isoforms (Notch1 through Notch4) are known to bind Delta-like or Jagged family ligands. This interaction triggers a series of proteolytic events leading to the active fragment Notch Intracellular Domain (Notch ICD), which then acts in the nucleus, where it associates with binding partners and transcriptional activators [23]. Several components of the Notch pathway are highly expressed at the invasive margins of tumors, which express EMT markers such as vimentin, suggesting an important role for the Notch pathway in the regulation of EMT [24]. Notch acts via transcriptional regulation of ZEB, Snail, and Slug, which repress expression of E-cadherin and induce expression of vimentin and fibronectin [23,24,25].

There is crosstalk between the Notch and TGFβ pathway that occurs via SMADs. As described above, the SMAD family of proteins are integral to TGFβ signaling. They have also been shown to associate with Notch-ICD. This affects the expression of genes downstream of both Notch and TGFβ that are required for mesenchymal differentiation, a key component in EMT [26]. Silencing components of the Notch pathway have also been shown to prevent TGFβ-induced EMT [27].

### 3.4. Tyrosine Kinase Pathways

Mitogenic growth factors also play a role in the regulation of EMT. The binding of these growth factors causes their receptors to dimerize and induces the activation of both receptor and non-receptor tyrosine kinases (TKs). This enables the activation of several pathways—including the MAPK, JAK-STAT, and phosphatidylinositol 3-kinase-Akt (PI3-Akt) pathways. All of these have been implicated in EMT, and are involved in cell growth, proliferation, and migration [28]. PI3K/Akt has also been shown to play an important role in the regulation of transcriptionally active β-catenin, a key molecule in the previously discussed *wnt* signaling pathway [29]. Inhibition of TKs is a growing field of study in cancer therapeutics, with multiple inhibitors currently under investigation [30].

Inhibition of fibroblast growth factor (FGF), a mitogenic growth factor that participates in the induction of EMT via activation of the MAPK, induces the reverse process MET in vitro and delays tumor growth in vivo [31]. One isoform, FGF2, has been associated with reduced overall survival in several carcinoma types if overexpressed [31].

The binding of epidermal growth factor (EGF) to its receptor leads to activation of MAPK pathway and decreased expression of E-cadherin [32]. EGF also activates the JAK2-STAT3 pathway, which leads to EMT activation via Twist [33]. Additionally, EGF has been shown to induce EMT via TGFβ signaling and regulation of Snail [34] and phosphorylation of SMAD2/SMAD3 [35]. 

The activation of Akt, or Protein Kinase B, has been shown to upregulate the phosphorylation of Twist1 and inhibit apoptosis [36], and the inhibition of Akt has been shown to induce MET [37]. For example, hepatocyte growth factor (HGF) has been shown to activate EMT [38], which can be reversed via inhibition of the PI3K/Akt pathway. HGF was found to enhance tumor progression and metastasis of hepatocellular carcinoma in association with the c-MET receptor tyrosine kinase [39], a known activator of PI3K/Akt.

## 4. EMT in OS

As a mesenchymal cancer, the importance of EMT in OS has been disputed [40,41]. In fact, an early investigation including 4 clinical osteosarcoma samples by Sato et al. found there was no detectable E-cadherin expression in these cells [42], suggesting that the repression/downregulation of E-cadherin—a classically described step in EMT—would not be possible. In contrast, Yin et al. found that 20.6% of OS tissue samples expressed E-cadherin and those that did were less likely to metastasize, whereas the expression of Twist was significantly related to metastases and poorer overall survival [43]. The promotion of EMT in OS characterized by increased migration and invasion in vitro has been shown to be mediated via upregulation of Snail [44,45,46,47,48,49,50,51,52,53,54,55,56,57], Slug [58,59], Twist [60,61,62,63], and ZEB [64,65,66,67].

The following sections give an overview of studies that have examined the roles of different EMT regulatory molecules in OS. All of these were found to affect the expression of EMT-related cell-markers and are correlated closely with EMT-associated cellular features such as increased migration and invasion. Many also showed a link between their proposed EMT-regulatory molecule and OS metastases in vivo in animal models. Taken together, these results suggest that EMT does play a role in osteosarcoma and is associated with a more aggressive tumor phenotype. However, the term “transition” is not ideally suited to sarcoma cancers, and EMT may be better thought of as a set of pathways utilized to maintain and promote the existing mesenchymal phenotype.

Sannino et al. posited a possible hybrid phenotype in sarcoma tumors cells, utilizing the EMT and MET pathways to acquire both mesenchymal and epithelial characteristics that favor the initiation and establishment of distant metastases [40]. 

The highlighted pathways important in EMT regulation have all been shown to have a role in OS. TGFβs promote EMT and metastases in OS [68], and TGFβ inhibition has been shown to decrease EMT in OS [58,69,70,71,72,73,74]. Chen et al. also identified that estrogen-related receptor α (ERRα) upregulates TGF-β-mediated EMT in two OS cell lines [50]. Others have highlighted roles for MAPK [63,75,76,77] and JAK/STAT [52,78,79,80,81,82].

Notch signaling promotes proliferation, migration, and invasion of OS cells, and Notch overexpression increased tumor growth in vivo [83,84,85,86]. Notch inhibition reduced chemo-resistance in OS in vitro [87,88]. *Wnt* signaling has also been shown to mediate EMT in OS [49,89,90,91,92,93,94,95,96,97,98,99,100,101,102,103,104]. It has been proposed that *wnt* signaling is particularly important in the pathogenesis of OS cancer stem cells [105]. 

TKs are of particular interest in OS, and multiple different TK proteins have been associated with aggressive cellular phenotypes. Many studies have demonstrated a regulatory role in OS for the downstream TK pathways PI3K/Akt [106,107,108,109,110,111,112,113,114] and RAS/RAF/MEK/ERK [61,115,116,117,118]. Multiple TK inhibitors have been a part of recently completed or ongoing clinical trials in the treatment of OS, including Apatinib, Axitinib, Cabozantinib, Cediranib, Crizotinib, Dasatinib, Imatinib, Pazopanib, Regorafenib, Sorafenib, and Sunitinib [119,120].

## 5. Regulation of EMT in OS—Proteins

As a complex and multi-faceted process, several proteins have been implicated in EMT regulation in OS [44,48,49,53,54,55,61,62,64,78,89,91,93,94,97,100,106,107,108,109,112,113,114,116,121,122,123,124,125,126,127,128,129,130,131,132,133,134,135,136,137,138,139,140,141,142,143,144,145,146,147,148,149,150,151], and these are summarized in Table 1 and Table 2. These proteins were found to either promote [44,48,53,54,61,64,78,89,93,94,97,100,106,107,108,109,112,113,114,116,121,122,123,124,125,126,127,128,129,130,131,132,133,134,135,136,137,138,139,140,141,142,143,144,145,146] or inhibit [49,55,62,91,147,148,149,150,151] EMT in vitro, and the majority were found to be correspondingly upregulated or downregulated in clinical OS tissue samples and/or established cell lines compared to normal controls. Each group of authors found a significant correlation between the studied protein and the levels of EMT-related proteins, such as E-cadherin, N-cadherin, and vimentin. They also reported a significant effect on aggressive cellular characteristics, such as migration and invasion ability in vitro. Where noted, the results were confirmed in vivo with mouse xenograft experiments.

A detailed review of the individual proteins investigated for their regulatory role in EMT of OS cells is outside of the scope of this review. Generally, their endogenous functions can be grouped into the following families: cell cycle regulation, immunity/inflammatory, cell signaling, cell structure, and metabolism. Each of these categories has a logical impact on EMT and/or cancer cell behavior. 

Changes to cell differentiation and cell cycle regulation are recognized mechanisms by which normal cells can become cancerous. We identified 22 proteins with a regulatory role in EMT in OS whose endogenous functions impact these processes [53,54,62,64,78,91,97,100,112,116,126,130,131,139,140,141,142,143,144,145,146,151]. This group can be represented by several ubiquitin ligases [142,143] and deubiquitinases [97,145,146] that are known to target proteins critical for cell growth, proliferation, and differentiation. These were all found to be upregulated in clinical samples of osteosarcoma, and the overexpression or inhibition of these proteins was found to correlate with markers for EMT and OS cell proliferation.

The importance of immunity and inflammation on cancer progression is widely recognized [152], and these systems have also been implicated in the regulation of EMT [153]. Twelve of the EMT regulatory proteins described in Table 1 and Table 2 function as part of immunity and inflammation [44,48,107,109,114,123,124,127,132,135,137,150], many of which can be designated as “pro-inflammatory” proteins [107,114,123,124], and others that function as part of the development and activation of immune cells [44,48,109,127,137]. For example, the Programmed Death Ligand 2 (PD-L2) protein is a ligand for the Programmed Death 1 (PD-1) receptor, which is protective against T cell-mediated death in conjunction with tumor-associated macrophages [154]. Ren et al. found that PD-L2 knockdown decreased EMT and inhibited migration, invasion, and colony formation of OS cells in vitro, and reduced OS metastases in vivo in a mouse model [137].

**Table 1 biomolecules-13-00398-t001:** Effects of highly expressed proteins on EMT in OS.

Protein	Increased Levels in Clinical Sample	Promoted EMT	Promoted Cell Migration/Invasion	Promoted In Vivo Tumor Growth	Promoted In Vivo Metastasis	Endogenous Function
ACTL6A [121]	Yes					Structure
AIM2 [114]	No					Immune
BMP-2 [93]	No					Cell Signaling
Calponin 3 [122]	Yes					Structure
Cathepsin E [123]	Yes					Immune
COPS3 [116]	Yes			ns		Cell Cycle Regulation
COX2 [107]	No					Inflammation
CPE-ΔN [94]	No					Cell Signaling
cPLA_2_a [124]	Yes					Inflammation
CPXM2 [125]	Yes					Cell Signaling
Cul4A [126]	Yes					Inflammation
CXCR6 [127]	Yes					Immune
Cyr61 [61,128]	No					Cell Signaling
E2F1 [78]	Yes					Cell Cycle Regulation
EPB41L3 [129]	Yes	↓	↓			Structure
Fibulin-3 [89]	Yes					Structure
Fibulin-4 [106]	Yes					Structure
HOXB7 [130]	Yes					Cell Cycle Regulation
HuR [131]	Yes					Cell Cycle Regulation
ICSBP [44]	Yes					Immune
IL-33 [109]	No					Inflammation
MAGL [132]	Yes					Inflammation
Metadherin [133]	No					Cell Signaling
NETO2 [113]	Yes					Cell Signaling
OLR1 [134]	Yes					Cell Signaling
P2X7 [108]	Yes					Cell Signaling
PADI4 [135]	Yes					Inflammation
PDGFRβ [112]	No					Cell Cycle Regulation
PD-L2 [137]	Yes					Inflammation
PGI [136]	No					Metabolism
RIPK4 [100]	Yes					Cell Cycle Regulation
SenP1 [138]	No					Cell Signaling
SENP3 [139]	Yes					Cell Cycle Regulation
SIRT1 [64]	Yes					Cell Cycle Regulation
SOX3 [53]	Yes					Cell Cycle Regulation
SOX5 [54]	Yes					Cell Cycle Regulation
ST6Gal-1 [140]	No					Cell Cycle Regulation
TANK1 [141]	No					Cell Cycle Regulation
Tim-3 [48]	Yes					Immune
TRIM29 [142]	Yes					Cell Cycle Regulation
TRIM66 [143]	Yes					Cell Cycle Regulation
UHRF1 [144]	No					Cell Cycle Regulation
USP7 [97]	Yes					Cell Cycle Regulation
USP17 [145]	Yes					Cell Cycle Regulation
USP22 [146]	Yes					Cell Cycle Regulation

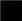
: Significant association; ns: No significant association; ↓: Inverse association; 
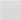
: Not studied/reported.

**Table 2 biomolecules-13-00398-t002:** Effects of poorly expressed proteins in OS.

Protein	Decreased Levels in Clinical Sample	Inhibited EMT	Inhibited Cell Migration/Invasion	Inhibited In Vivo Tumor Growth	Inhibited In Vivo Metastasis	Endogenous Function
ARID1a [147]	Yes					Cell Signaling
Ezrin [148]	No					Structure
FTL [149]	Yes					Metabolism
GPER [55]	Yes					Cell Signaling
LAIR-1 [150]	Yes					Immune
RASSF4 [91]	Yes					Cell Cycle Regulation
SOX6 [62]	Yes					Cell Cycle Regulation
TSSC3 [49]	Yes					Cell Cycle Regulation
WWOX [151]	Yes					Cell Cycle Regulation

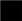
: Significant association; 
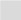
: Not studied/reported.

Many of the described proteins are implicated in cell signaling [55,93,94,108,113,126,129,134,135,139,148]. In addition to cell–cell interactions, this broad grouping includes the regulation of multiple cell processes that affect multiple other pathways and functions, including but not limited to cell cycle regulation, inflammation, immunity, and metabolism.

Finally, a subset of the proteins associated with EMT in OS are either structural proteins or regulate cell structure via interaction with the cytoskeleton [89,106,121,122,129,148]. This is perhaps the simplest and most logical grouping given the key morphological changes that take place during the EMT transformation, as depicted in Figure 1. Interestingly, Yuan et al. found that Erythrocyte Membrane Protein Band 4.1-like 3 (EPB41L3)—a cytoskeletal protein involved in cytoskeletal rearrangement, intracellular transport, and signal transduction—was increased in OS tissues and cell lines but was associated with an inhibition of EMT, migration, invasion, and cell viability in OS cell culture [129]. This pattern of expression was opposite to all of the other proteins impacting EMT in OS identified in this review.

When reported, the EMT pathways most implicated in these studies were *wnt* and PI3K/Akt. The nuclear localization and, therefore, transcriptional activity of the *wnt*/β-catenin pathway has also been shown to be regulated by PI3K [30], suggesting overlap in these EMT control mechanisms. The most frequently identified downstream target was Snail, which is known to promote EMT by suppressing E-cadherin expression [155], and further upregulates *wnt* signaling and EMT [20].

## 6. Regulation of EMT in OS—Non-Coding Ribonucleic Acids

Another key group of regulatory factors of EMT/MET in OS are non-coding ribonucleic acids (ncRNAs). These molecules have many forms and functions [156], one of which is gene regulation. Typically identified through queries to the Gene Expression Omnibus (GEO), the differential expression of multiple separate long non-coding RNAs (lncRNAs) [157], microRNAs (miRNAs) [158], circular RNAs (circRNAs) [159,160], and pseudogenes [161] have been found to relate to OS prognosis [162]. Table 3 and Table 4 summarize ncRNAs implicated in OS EMT regulation. Again, they were found to have a role in either promoting [57,63,66,76,85,86,90,95,98,99,101,103,109,163,164,165,166,167,168,169,170,171,172,173,174,175,176,177,178,179,180,181,182,183,184,185,186,187,188,189,190,191,192,193,194,195,196] or inhibiting [47,49,51,65,77,92,96,102,104,111,197,198,199,200,201,202,203,204,205,206,207,208,209,210,211,212,213,214,215,216,217,218,219,220,221,222,223,224,225] EMT and invasive cellular behaviors of OS cells in vitro, and there was significant overlap in the affected pathways and ultimate downstream targets. Very frequently there are multiple non-coding RNAs involved in the same pathway as they can also regulate other nucleic acids. 

Unlike the pattern observed in the majority of these findings, Yuan et al. found that although erythrocyte membrane protein band 4.1-like 3 (EPB41L3) was upregulated in OS cell lines and clinical tissue samples, knockdown of EPB41L3 significantly increased the migration and invasion capacity of the investigated cell lines despite decreased cell viability [130]. The findings were similarly mixed for lncRNA NKILA [201] and miR-let-7d [225]. These studies highlight the complexity of EMT regulation in OS and suggest that it is only one possible factor relating to tumor behavior and prognosis.

## 7. Regulation of EMT in OS—The Tumor Microenvironment

There has been increased recognition of the importance of the tumor microenvironment on various cellular functions and characteristics. This is the three-dimensional structure surrounding tumor cells and comprises immune cells, vascular network, and extra-cellular matrix (ECM), among other components. The tumor microenvironment is unique not only for different cancer types but also for individual patients, and it is influenced by multiple factors, including patient sex and presence of metastases [226]. A better understanding of the interactions within the tumor micro-environment is expected to lead to the development of personalized treatments targeted at individual patients’ tumors.

Han et al. found that the presence of tumor-associated macrophages (TAMS) and the expression of the inflammatory marker cyclo-oxygenase 2 (COX2) correlated with OS metastases in clinical samples [80]. They also found co-culture of OS cells with TAMS promoted EMT and aggressive cellular features in vitro, which was reversible by COX2 inhibition. Additionally, COX2 inhibition reduced pulmonary metastases in vivo in a murine model [80]. Ling et al. found that Von Willebrand Factor (VWF)—which is secreted by the endothelial cells lining blood vessels—promoted EMT in vitro following OS and endothelial cell co-culture, as well as tumor growth and metastasis in vivo in a mouse model [227].

In addition to the cellular and biochemical makeup of the tumor microenvironment, the biomechanical properties of the ECM may also play a role in regulating EMT. Dai et al. developed a three-dimensional cell culture model with varying degrees of ECM stiffness [228]. This may be of particular relevance when evaluating OS tumors that exist in the bone—a relatively rigid environment—but eventually expand into the surrounding soft tissues, which are substantially less rigid. It may also account for some of the differences in OS metastatic patterns as more than 85% of metastatic OS occurs in the lungs, a soft tissue, compared to only 21% that occurs in the bone [229].

## 8. Targeting EMT in Osteosarcoma

Given its close association with aggressive and metastatic OS, EMT is a natural target for OS treatment. Treatments targeting EMT in OS include known medications, hormones, novel small molecules, and herbal medicines. The currently recommended chemotherapy regimen for OS includes doxorubicin, cisplatin, and high-dose methotrexate. At low doses, cisplatin has been shown to promote EMT, migration, invasion, and in vivo tumor growth [87]. However, these findings were reversible with inhibition of the EMT-related Notch pathway. EMT is recognized as an important factor contributing to chemotherapy resistance in multiple cancers [230], which was shown in OS by Ding et al., who found that OS cells induced to be resistant to methotrexate exhibited higher levels of EMT proteins and greater migration and invasion [231].

Other drugs that have not traditionally been used to treat osteosarcoma clinically but have been shown to inhibit EMT in OS in vitro include the cholesterol medication lovastatin [232] and zolendronate, a bisphosphonate medication used in the treatment of osteoporosis and other metabolic bone disorders [233,234]. In addition to the suppression of EMT, migration, and invasion, Kim et al. showed that OS cells and orthotopic tumors in mice had increased radiation-sensitivity following treatment with zolendronate, and this combination therapy was more effective than either treatment on its own [234]. 

Several hormone therapies have also been investigated for their effect on EMT in OS. These include estrogen, which inhibited EMT and promoted apoptosis of OS cells at high doses [235]. Treatment with irisin, a hormone derived from skeletal muscle, also suppressed OS EMT, cell proliferation, migration, and invasion [52]. Melatonin, a sleep-related hormone that is widely commercially available, has been shown to inhibit EMT [56,236,237] and OS cell migration and invasion in vitro, with additional results in vivo showing reduced metastasis in mice [237]. In contrast, treatment of OS cells with visfatin [45], a metabolic peptide first identified in visceral fat, induced EMT and increased cell migration and invasion. While these results do not suggest a direct role for visfatin in treatment of OS, further studies could examine the potential therapeutic effects of visfatin regulation.

Newer therapies with peptides and other small molecules allow for targeting more specific biologic functions, often associated with receptor inhibition. Inhibition of CXCR4 with Peptide R inhibited EMT, cell migration, and invasion in OS cells, and was thought to have the potential for less toxicity than existing CXCR4 inhibitors [238]. Similar suppression of EMT, inhibition of cell migration/invasion, and reduced tumorigenesis in vivo was observed with inhibition of vascular endothelial growth factor receptor-2 (VEGFR2) by Apatinib [239], Krüppel-like factor 5 (KFL5), and early growth response gene 1 (EGR1) by ML264 [81], and TGF-β by RepSox [70]. The 4′-aminochalcones D14 and D15 were found to inhibit EMT, cell migration, and invasion through upregulation of p53 [240].

The investigation of traditional and herbal medicines and their derivatives (both natural and synthetic) is a growing area of interest. The effect on EMT in OS has been studied for a number of these compounds [46,58,69,71,72,73,74,75,82,115,118,241,242,243,244,245,246,247,248,249], with results summarized in Table 5. While the majority of these inhibited both EMT and aggressive cellular characteristics, such as migration and invasion, Jiang et al. found that triptolide, a compound found in the vine *Tripterygium wilfordii*, increased EMT in OS cells in vitro but inhibited proliferation and invasion [249].

A non-pharmacological treatment targeting EMT in OS was also reported. Tumor-treating electrical field (TTEF) was reported to suppress EMT, cell migration, invasion, and angiogenesis of OS cells in culture via potential effects on VEGF and matrix metalloproteinase 2 (MMP2) [250].

## 9. Conclusions

EMT has significant implications in OS, despite its mesenchymal origin. Multiple studies have correlated changes in EMT with a more aggressive OS phenotype, both in vitro and in vivo. More than 100 proteins and non-coding nucleic acids have been identified as having a potential regulatory role in the OS EMT/MET pathways, and these may prove to be viable therapeutic targets and/or prognostic factors. These results should be interpreted with caution. While many of the studies discussed in this review confirmed the presence of their specific molecule of interest in clinical samples, most of the cell culture and animal studies were performed with only a handful of established cell lines. The majority of OS samples do not exhibit any E-cadherin and would therefore not experience a significant change secondary to E-cadherin suppression, a key process in EMT. It is possible that the cell lines most frequently utilized for these investigations are in the minority that do express E-cadherin and therefore exaggerate the EMT effect. Unfortunately, as OS is a rare cancer, any findings such as these are difficult to generalize. However, a role for EMT/MET has clearly been shown in cell culture and may well be a viable therapeutic target. Further work in additional cell lines or primary cell culture would help to confirm the findings outlined in this review.

## Figures and Tables

**Figure 1 biomolecules-13-00398-f001:**
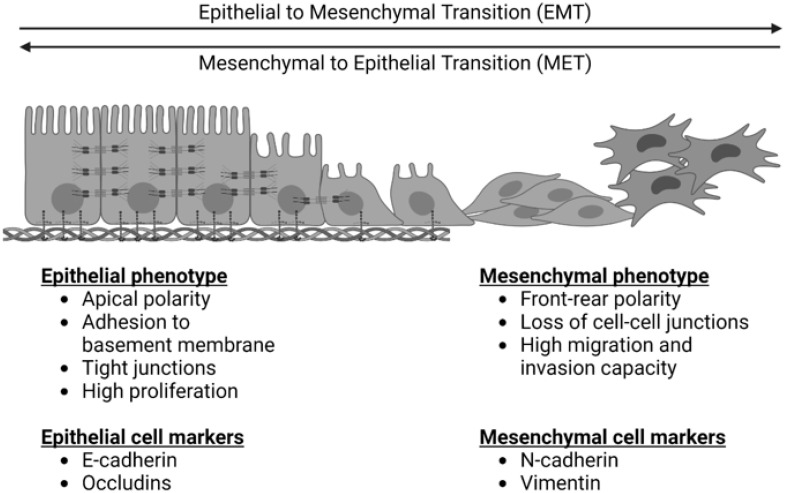
The epithelial to mesenchymal transition (EMT) and the reverse process of the mesenchymal to epithelial transition (MET). EMT is characterized by a loss of epithelial cell markers, an increase in mesenchymal cell markers, a loss of apical cell polarity, a loss of tight cell junctions, and an increased capacity for cell migration and invasion. MET is characterized by a loss of mesenchymal cell markers, an increase in epithelial cell markers, increased apical cell polarity, tight junctions, adherence to a basement membrane, and increased cell proliferation.

**Figure 2 biomolecules-13-00398-f002:**
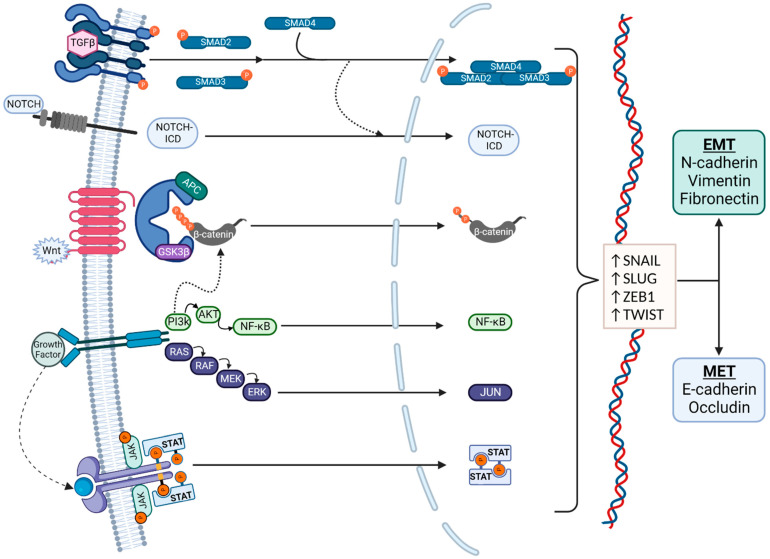
Signaling pathways in EMT. EMT regulation is complex and affected by multiple pathways, which also interact with each other. Regulation is typically via the Transforming Growth Factor β (TGFβ)/SMAD, Notch, canonical *wnt*, phosphoinositide 3-kinase (PI3K)/Akt, RAS/RAF, and JAK/STAT pathways. The transcription factors that mediate EMT are primarily Snail, Slug, ZEB1, and TWIST. EMT is characterized by an increased production of N-cadherin, vimentin, and fibronectin, and MET is characterized by an increased production of E-cadherin and Occludin.

**Table 3 biomolecules-13-00398-t003:** Effects of highly expressed non-coding ribonucleic acids.

Ribonucleic Acid	Increased Levels in Clinical Sample	Promoted EMT	Promoted Cell Migration/Invasion	Promoted In Vivo Tumor Growth	Promoted In Vivo Metastasis	Associated Pathways/Targets
circ-FOXM1 [103]	No					miR-320a, *wnt*
circ-PRKAR1B [163]	No					miR-361-3p, FZD4
LINC00319 [164]	Yes					miR-455-3p, NFIB
LINC00324 [165]	Yes					WDR66, HuR
LINC00460 [166]	No					miR-1224-5p, FADS1
LINC02381 [167]	Yes					miR-503-5p, CDCA4
lncRNA AFAP1-AS1 [63]	Yes					Rho, ROCK, p38
lncRNA BCRT1 [168]	Yes					miR-1303, FGF7
lncRNA CASC15 [101]	Yes					*wnt*/β-catenin
lncRNA CCAT2 [169]	Yes					LATS2, c-Myc
lncRNA CRNDE [85,98]	Yes					Notch1, SP1, *wnt*/β-catenin
lncRNA DDX11-AS1 [170]	No					miR-873-5p, IGF2BP2
lncRNA FAL1 [171]	Yes					GSK-3β
lncRNA GHET1 [99,172]	Yes					Ki67, *wnt*/β-catenin
lncRNA HCP5 [173]	No					SP1
lncRNA HNF1A-AS1 [174]	Yes					
lncRNA HIF1A-AS2 [175]	Yes					miR-33b-5p, SIRT6
lncRNA HOXA-AS2 [176]	Yes					miR-502c-3p
lncRNA LMCD1-AS1 [177]	Yes					miR-106b-5p, SP1
lncRNA miR210HG [178]	Yes					miR-503
lncRNA MNX1-AS1 [179]	Yes					Snail
lncRNA MSC-AS1 [180]	Yes					miR-142, CDK6, PI3K/Akt
lncRNA NEAT1 [181]	Yes					miR-186-5p, HIF-1α
lncRNA PGM5-AS1 [182]	Yes					miR-140-5p, FBN1
lncRNA PVT1 [183]	Yes					
lncRNA RUSC1-AS1 [184]	Yes					miR-340-5p, PI3K/Akt
lncRNA SNHG1 [185]	Yes					miRNA-101-3p, ROCK1, PI3K/Akt
lncRNA SNHG4 [186]	Yes					miR-377-3p
lncRNA SNHG7 [86]	Yes					MiR-34a, Notch-1, BCL-2, CDK6, SMAD4
lncRNA SNHG20 [187]	Yes					
lncRNA SPRY4-IT1 [47,66]	No					miR-101
lncRNA TUG1 [90,188]	Yes					miR-144-3p, miR-143-5p, EZH2, HIF-1α, *wnt*
lncRNA XIST [57]	Yes					miR-153, SNAI1
miR-17-5p [189]	Yes					SRCIN1
miR-19 [76]	Yes					SPRED2, ERK/MAPK
miR-31-5p [95]	Yes					AXIN1, *wnt*/β-catenin
miR-93 [190]	Yes					TIMP2
miR-130a [191]	Yes					PTEN
miR-135b [192]	No					TAZ
miR-155 [193]	No					TNFa, TP53INP1
miR-196a [194]	Yes					HOXA5
miR-199b-5p [195]	Yes					HER2
miR-210-5p [110]	Yes					PIK3R5, Akt
Pseudogene MSTO2P [196]	Yes					PD-L1

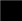
: Significant association; 
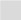
: Not studied/reported.

**Table 4 biomolecules-13-00398-t004:** Effects of poorly expressed non-coding ribonucleic acids.

Ribonucleic Acid	Decreased Levels in Clinical Samples	Inhibited EMT	Inhibited Cell Migration/Invasion	Inhibited In Vivo Tumor Growth	Inhibited In Vivo Metastasis	Associated Pathways/Targets
lncRNA FER1L4 [197,198]	Yes					miR-18a-5p, PI3K/Akt
lncRNA GAS5 [199]	Yes					miR-221, ARHI
lncRNA MEG3 [200]	Yes					miR-361-5p, FoxM1
lncRNA NKILA [201]	Yes	↑				NFκB, Snail
lncRNA TUSC8 [202]	Yes					miR-197-3p, EHD2
miR-7 [203]	Yes					IGF1R
miR-16 [204]	Yes					RAB23
miR-25 [205]	Yes					SOX4
miR-29a [206]	Yes					SOCS1/NFκB, DNMT3B
miR-107 [92]	Yes					*wnt*/β-catenin
miR-125a-5p [207]	Yes					MMP11
miR-128 [208]	Yes					Integrin A2
miR-132 [209]	No					SOX4
miR-139-5p [210]	Yes					DNMT1
miR-140-3p [211]	Yes					TRAF6, TGFB
miR-145 [51]	Yes					Snail
miR-181c [212]	Yes					SMAD7, TGFB
miR-203 [213]	Yes					RAB22A
miR-331-3p [104]	Yes					MGAT1, Bcl/Bax, *wnt*/β-catenin
miR-342-5p [96]	Yes					*wnt*/β-catenin
miR-363 [214,215]	Yes					PDZD2, NOB1
miR-377-3p [186]	Yes					CuL1, *wnt*/β-catenin
miR-382 [216]	Yes					YB-1
miR-384 [217]	Yes					MECP2, IGFBP3
miR-449a [111]	Yes					EZH2, PI3K/Akt
miR-486 [218]	Yes					PIM1
miR-488 [219]	Yes					AQP3
miR-489 [220]	Yes					NAA10
miR-499a [221]	Yes					TGFβ, EGFR, Akt, SHKBP1
miR-503 [222]	No					c-myc
miR-506-3p [223]	No					SPHK1, LC3II/I
miR-708-5p [65]	No					ZEB1
miR-761 [224]	Yes					ALDH1B1, TGFB
miR-765 [117]	Yes					MTUS, ERK
miR-CT3 [77]	Yes					p38/MAPK
miR-let-7d [225]	No		↑			CCND2, E2F2

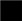
: Significant association; ↑: Inverse association; 
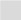
:Not studied/reported.

**Table 5 biomolecules-13-00398-t005:** Traditional and herbal medicine effects on EMT in OS.

Compound	Inhibits EMT	Inhibits Cell Migration/Invasion	Inhibits In Vivo Tumor Growth	Inhibits In Vivo Metastasis	Associated Pathways/Targets
3’hydroxyflavone [115]					MEK/ERK
Baicalin [74,118]					ERK, TGF-β
Berberine [241,242]					EZH2, Rad51
Chimaphilin [73]					PI3K/Akt, ERK, TGF-β
Cinnamomum cassia extract [69]					TGF-β
Dehydroandrogranpholide [243]					SATB2
Delphinidin [75]					ERK, MAPK
Gamabufotalin [71]					PI3K/Akt, TGF-β
Glaucocalyxin A [72]					TGF-β, Smad
Magnoflorine [244]					miR-410-3p, HMGB1, NF-κB
Nitidine Chloride [46]					Akt, GSK-3β, Snail
Oridonin [58]					TGF-β, Smad, Snail
Piperlongumine [82]					miR-30d-5p, SOCS3, JAK2/STAT3
Polyphillin I [245]					NF-κB, c-Myc
Rosmarinic acid [246]					DJ-1, PI3K/Akt
Salvia 13landestine extract [247]					Akt/PKB
Sauchinone [248]					Sonic hedgehog
Triptolide [249]	↑				miR-181a, PTEN

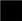
: Significant association; ↑: Inverse association; 
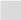
: Not studied/reported.

## Data Availability

Not applicable.

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
