# Peer review of "Regulation of the Epithelial to Mesenchymal Transition in Osteosarcoma"

_biomolecules, 2023, doi:10.3390/biom13020398_

Round 1

Reviewer 1 Report

Overall, this is a clear, concise, and well-written review paper.

1)      P3, Line 54 - The authors need to mention about the the role of EMT in the tumorigenesis of different cancer. 

2)      P3, Line 75 - The author need to mention about the main events in EMT – figure is required.

3)      P5, Line 103 – The author need to mention about the molecular mechanisms of TGF beta induced EMT.

4)      P6, Line 157 – The author need to mention about the downstream targets of Notch pathway signaling and their target for therapy.

5)      P6, Line 172 – registered clinical trials for Tyrosine kinase pathway for osteosarcoma treatment such as Apatinib, Axitinib and cabozantinib need to be mentioned.

6)      P6, Line 200 – role of EMT mediator transcription factors in osteosarcoma like twist, Zeb and snail need to be mentioned in detail.

7)      References: need to refer latest papers.

Author Response

The authors thank Reviewer 1 for their helpful comments. We hope that our responses to the reviewer’s comments are acceptable.

Overall, this is a clear, concise, and well-written review paper.

1)      P3, Line 54 - The authors need to mention about the role of EMT in the tumorigenesis of different cancer.

                Response: A statement has now been added to address this comment.

2)      P3, Line 75 - The author need to mention about the main events in EMT – figure is required.

Response: Additional details have been added to the text and adjustments have been made to better depict the changes in Figure 1.

3)      P5, Line 103 – The author need to mention about the molecular mechanisms of TGF beta induced EMT.

Response: This section has been reworded to attempt to make the role of TGF signalling in EMT clearer without repeating a large portion of the description of the events of EMT.

4)      P6, Line 157 – The author need to mention about the downstream targets of Notch pathway signaling and their target for therapy.

                Response: Information has now been added to address this comment.

5)      P6, Line 172 – registered clinical trials for Tyrosine kinase pathway for osteosarcoma treatment such as Apatinib, Axitinib and cabozantinib need to be mentioned.

                Response: Information has been added in the later section EMT in OS to address this comment.

6)      P6, Line 200 – role of EMT mediator transcription factors in osteosarcoma like twist, Zeb and snail need to be mentioned in detail.

                Response: Appropriate citations have been added to highlight the roles of these factors specifically in OS.

7)      References: need to refer latest papers.

Response: References were updated where possible. We aimed to capture as many articles as possible that studied EMT in osteosarcoma, many of which were in the past five years, but some are older. Many of the references for the background information were updated but older primary sources to support certain statements were maintained.

Reviewer 2 Report

Kristin Hinton at al. reviewed the regulation of the epithelial to mesenchymal transition (EMT) in osteosarcoma (OS). This is a good choice of topic. EMT is highly complex and regulated via multiple signalling pathways, but as a mini-review article, the regulation of EMT in OS should be well introduced. The overall structure of the article is relatively clear, but there are still some issues that need to be added.

A few major revisions are list below:

1. The author introduced the regulation of EMT via the canonical Wnt/β-catenin pathway, the NOTCH pathway, the TGFβ/SMAD pathway, the PI3K/Akt pathway, the p38 MAPK pathway, the RAS/RAF/MEK/ERK axis, and the JAK/STAT pathway. However, the regulation of EMT in OS via these signaling pathways was not well introduced in the part of EMT IN OS. The part needs numerous references to demonstrate the regulation of EMT in OS via these theses signaling pathways, rather than theses signaling pathways only affect EMT.

2. In Figure 2, the author should describe the effect of these signaling pathways on the levels of ZEB, SNAIL, and TWIST in detail (up-regulation, down-regulation, or no effect). It will be better understand the role of these signaling pathways in EMT regulation in the part of EMT SIGNALLING PATHWAYS.

3. As a review, the references are obviously outdated; please use the last 3 years of literature if possible. There are only about 76 articles (2020-2022) in the 250 references.

4. In line 83, abbreviations need full names the first time they appear, such as ZEB, SNAIL, and TWIST.

Author Response

The authors thank Reviewer 2 for their helpful comments. We hope that our responses to the reviewer’s comments are acceptable.

Kristin Hinton at al. reviewed the regulation of the epithelial to mesenchymal transition (EMT) in osteosarcoma (OS). This is a good choice of topic. EMT is highly complex and regulated via multiple signalling pathways, but as a mini-review article, the regulation of EMT in OS should be well introduced. The overall structure of the article is relatively clear, but there are still some issues that need to be added.

A few major revisions are list below:

  1. The author introduced the regulation of EMT via the canonical Wnt/β-catenin pathway, the NOTCH pathway, the TGFβ/SMAD pathway, the PI3K/Akt pathway, the p38 MAPK pathway, the RAS/RAF/MEK/ERK axis, and the JAK/STAT pathway. However, the regulation of EMT in OS via these signaling pathways was not well introduced in the part of EMT IN OS. The part needs numerous references to demonstrate the regulation of EMT in OS via these theses signaling pathways, rather than theses signaling pathways only affect EMT.

Response: References to address this comment have been added to demonstrate the regulation of these pathways in OS EMT.

  1. In Figure 2, the author should describe the effect of these signaling pathways on the levels of ZEB, SNAIL, and TWIST in detail (up-regulation, down-regulation, or no effect). It will be better understand the role of these signaling pathways in EMT regulation in the part of EMT SIGNALLING PATHWAYS.

                Response: This information has been added to Figure 2 to address this comment.

  1. As a review, the references are obviously outdated; please use the last 3 years of literature if possible. There are only about 76 articles (2020-2022) in the 250 references.

Response: References were updated where possible. We aimed to capture as many articles as possible that studied EMT in osteosarcoma, many of which were in the past five years, but some are older. Many of the references for the background information were updated but older primary sources to support certain statements were maintained.

  1. In line 83, abbreviations need full names the first time they appear, such as ZEB, SNAIL, and TWIST.

                Response: This information has now been added in the appropriate section.

Reviewer 3 Report

In this review article, Hinton et al have summarized potentially regulation of the epithelial to mesenchymal transition (EMT) in osteosarcoma.  In the introduction part, they review EMT in cancer and in signaling pathways. Later, they focus on EMT in osteosarcoma basically by listing proteins and “non-coding nucleic acids” that may have relation to EMT. Finally, the authors describe the role of tumor microenvironment in osteosarcoma and discuss potential anti-OS treatments. Although the manuscript contains some novel aspects of EMT and osteosarcoma, it is the opinion of the reviewer that this work is not suitable for publication in the current form.

Specific comments:

1, The introduction part of the review, describing EMT in cancer and signaling pathways in overall is too long. Readers interested in the general description of EMT can find dozens of excellent reviews on the field.

2, In page 4, authors write the following sentence: “The TGFβ signaling cascade involves three isoforms including two activins and bone morphogenic proteins (BMPs).” While TGFβ has really three different mammalian isoforms (TGF-β 1 to 3) these do not include activins and bone morphogenic proteins. These belong to the transforming growth factor beta (TGF-β) superfamily.

3, In page 6, the authors claim that “as a mesenchymal cancer, the importance of EMT in OS may be disputed.” To be honest, the reviewer have not identified any detailed mechanism by which the processes described in epithelial to mesenchymal transition relate to the pathogenesis of osteosarcoma. Instead, many EMT genes and proteins are listed and their changes in OS are described without any explanation why and how they are involved in OS.

4, In page 9, instead of “non-coding nucleic acids”, either in the title or in the text, I suggest using non-coding ribonucleic acids (ncRNAs).

Author Response

The authors thank Reviewer 3 for their helpful comments. We hope that our responses to the reviewer’s comments are acceptable.

In this review article, Hinton et al have summarized potentially regulation of the epithelial to mesenchymal transition (EMT) in osteosarcoma.  In the introduction part, they review EMT in cancer and in signaling pathways. Later, they focus on EMT in osteosarcoma basically by listing proteins and “non-coding nucleic acids” that may have relation to EMT. Finally, the authors describe the role of tumor microenvironment in osteosarcoma and discuss potential anti-OS treatments. Although the manuscript contains some novel aspects of EMT and osteosarcoma, it is the opinion of the reviewer that this work is not suitable for publication in the current form.

1, The introduction part of the review, describing EMT in cancer and signaling pathways in overall is too long. Readers interested in the general description of EMT can find dozens of excellent reviews on the field.

Response: We agree that this is not intended to be an exhaustive review of EMT in all cancers. However, we believe it is important to provide this context for readers, whose primary interest is osteosarcoma, but who may not be familiar with the events and regulation of EMT and its importance in cancer pathogenesis. We have removed some of the more detailed information from the general pathways section.

2, In page 4, authors write the following sentence: “The TGFβ signaling cascade involves three isoforms including two activins and bone morphogenic proteins (BMPs).” While TGFβ has really three different mammalian isoforms (TGF-β 1 to 3) these do not include activins and bone morphogenic proteins. These belong to the transforming growth factor beta (TGF-β) superfamily.

Response: The wording has been changed to reflect the classification of these proteins more clearly in accordance with this comment.

3, In page 6, the authors claim that “as a mesenchymal cancer, the importance of EMT in OS may be disputed.” To be honest, the reviewer have not identified any detailed mechanism by which the processes described in epithelial to mesenchymal transition relate to the pathogenesis of osteosarcoma. Instead, many EMT genes and proteins are listed and their changes in OS are described without any explanation why and how they are involved in OS.

Response: Additional justification has been added. The proteins/ncRNAs in OS that are highlighted in this review have specifically been shown to downregulate E-cadherin, upregulate N-cadherin and/or vimentin, increase cellular migration, invasion, and—in some cases—metastasis in vivo (or the reverse findings in the case of negative regulatory molecules) as outlined in the summary tables. While the detailed mechanism has not yet been completely elucidated, we believe these findings strongly support a role for EMT in OS.

4, In page 9, instead of “non-coding nucleic acids”, either in the title or in the text, I suggest using non-coding ribonucleic acids (ncRNAs).

Response: This change has been made. In the course of verifying this applied to each of the listed molecules, we identified a protein gene product that had been mis-classified as DNA (tumor suppressing STF cDNA 3, TSSC3)—this has been moved to the appropriate section.

Round 2

Reviewer 1 Report

authors has done the required corrections.

Reviewer 2 Report

No other suggestions

Reviewer 3 Report

I support publication of the manuscript in the current form.